# Ketogenic Diet Induced Shifts in the Gut Microbiome Associate with Changes to Inflammatory Cytokines and Brain-Related miRNAs in Children with Autism Spectrum Disorder

**DOI:** 10.3390/nu16101401

**Published:** 2024-05-07

**Authors:** Nina P. Allan, Brennan Y. Yamamoto, Braden P. Kunihiro, Chandler K. L. Nunokawa, Noelle C. Rubas, Riley K. Wells, Lesley Umeda, Krit Phankitnirundorn, Amada Torres, Rafael Peres, Emi Takahashi, Alika K. Maunakea

**Affiliations:** 1Department of Biochemistry, Anatomy, and Physiology, University of Hawai’i at Mānoa, Honolulu, HI 96822, USA; ninapa@hawaii.edu (N.P.A.); brennany@hawaii.edu (B.Y.Y.); bradenku@hawaii.edu (B.P.K.); cknunoka@hawaii.edu (C.K.L.N.); nrubas@hawaii.edu (N.C.R.); rkwells@hawaii.edu (R.K.W.); umedal@hawaii.edu (L.U.); kritphan@hawaii.edu (K.P.); torres91@hawaii.edu (A.T.); peres@hawaii.edu (R.P.); 2Molecular Biosciences and Bioengineering, College of Tropical Agriculture and Human Resources, University of Hawai’i at Manoa, Honolulu, HI 96822, USA; 3Department of Radiology, Harvard Medical School, Boston, MA 02115, USA; emi.oki@mgh.harvard.edu; 4Athinoula A. Martinos Center for Biomedical Imaging, Massachusetts General Hospital, Charlestown, MA 02129, USA

**Keywords:** ketogenic diet, autism spectrum disorder, inflammation, microbiome, butyrate, BDNF, miRNA

## Abstract

In this interventional pilot study, we investigated the effects of a modified ketogenic diet (KD) on children with autism spectrum disorder (ASD). We previously observed improved behavioral symptoms in this cohort following the KD; this trial was registered with Clinicaltrials.gov (NCT02477904). This report details the alterations observed in the microbiota, inflammation markers, and microRNAs of seven children following a KD for a duration of 4 months. Our analysis included blood and stool samples, collected before and after the KD. After 4 months follow up, we found that the KD led to decreased plasma levels of proinflammatory cytokines (IL-12p70 and IL-1b) and brain-derived neurotrophic factor (BDNF). Additionally, we observed changes in the gut microbiome, increased expression of butyrate kinase in the gut, and altered levels of BDNF-associated miRNAs in the plasma. These cohort findings suggest that the KD may positively influence ASD sociability, as previously observed, by reducing inflammation, reversing gut microbial dysbiosis, and impacting the BDNF pathway related to brain activity.

## 1. Introduction

Autism Spectrum Disorder (ASD) is a neurodevelopmental disorder characterized by difficulties in social interaction and communication and repetitive, restrictive behaviors [1]. Current prevalence rates of ASD among children are estimated to be over 1 in 100 globally and 1 in 36 in the United States [2].

ASD is linked to altered immune function and increased inflammation [3,4,5]. Proinflammatory cytokines are consistently elevated in ASD and correlated with symptom severity [6,7]. ASD children also exhibit different T cell activation patterns, suggesting immune dysregulation that could be behind observed disturbances in behavior and developmental functioning [8]. Additionally, gastrointestinal (GI) problems are common among individuals with ASD, implying disruptions in gut microbiome composition [9].

Multiple studies have linked abnormal gut flora to the varying severity of ASD symptoms, suggesting the involvement of the gut–brain axis in the development of ASD [10]. Treatments impacting the gut microbiome, such as the ketogenic diet (KD), which increases ketone bodies in the blood, have shown promise in improving outcomes for some ASD children [11,12]. Ketone bodies, like *β*-hydroxybutyrate, can cross the blood–brain barrier and may have neuroprotective effects, enhancing sociability in ASD [11,13]. However, the exact mechanism by which gut flora interact with hepatically derived ketone bodies remains less understood. Notably, the gut-produced short chain fatty acid butyrate shares similarities with *β*-hydroxybutyrate and has been considered as a potential supplement for ASD [14,15,16,17].

In this interventional pilot study, we previously observed improved behavioral symptoms in this cohort following the KD according to test results of the Autism Diagnostic Observation Schedule (ADOS-2) [13]. In the current analysis, we sought to understand how the modified KD might affect the relationships between the gut microbiome and inflammatory regulation. Previous reports identified metabolomic changes in the blood after a KD, including elevated *β*-hydroxybutyrate, which correlated with improved ASD-associated behaviors [13,18]. Herein, we report a potential connection between elevated butyrate kinase (*BUK*) expression levels in the gut, enhanced gut microbial diversity, and altered inflammatory profiles in children with ASD who underwent a KD, previously linked with behavioral improvements [13].

## 2. Materials and Methods

### 2.1. Recruitment and Participant Enrollment

Participants with ASD were recruited at Shriner’s Hospital for Children, Honolulu, as previously described [13]. The interventional follow-up study was approved by the University of Hawaii Committee on Human Studies, and participants’ guardians provided written informed consent. The ketogenic diet (KD) was modified by a Registered Dietician/Nurse to suit metabolic considerations for ASD and included medium-chain triglyceride (MCT) oil supplementation [19,20]. Given that many participants already practiced gluten restriction, this was applied to all participants. Total net daily carbohydrates were limited to 20–25 g. Protein needs were dependent on the weight and age of the child; up to twice the RDA requirements were allowed. All additional energy needs were provided by various fats. MCT was administered daily, using either coconut or pure MCT oil, to comprise 20% of energy needs. For the first month, caregivers tested for urine ketones twice daily, and once daily subsequently. The goal was to maintain a consistent state of ketosis, but no defined level of ketosis was targeted [13].

The trial was registered with Clinicaltrials.gov (NCT02477904). We obtained post-diet blood samples from 11 children and stool samples from 7 children, out of an original cohort of 47 children, due to personnel reasons (Figure 1). 

### 2.2. Sample Processing

#### Blood Sample Processing

Non-fasting blood samples were collected using venipuncture techniques between 9 and 10 am. The separation of plasma and peripheral blood mononuclear cells (PBMCs) was performed within 24 h of blood sample collection using PBMC isolation tubes (STEMCELL Technologies, Vancouver, BC, Canada). PBMCs were stored in liquid nitrogen for further applications. Plasma was stored at −80 °C prior to assays. 

### 2.3. RNA/DNA Extraction from Stool Samples

Participants were provided with a Fisherbrand™ Commode Specimen Collection System (FisherScientific, Waltham, MA, USA) to take home and asked to return their samples, preserved in RNAlater™ Stabilization Solution (ThermoFisher Scientific, Inc., Vilnius, Lithuania), within a week of blood collection. Samples were stored at −20 °C until ready for extraction. To extract stool DNA and RNA, we used a MagMAX Microbiome Ultra Nuclei Kit on a KingFisher Duo Prime Purification System following the manufacturer’s protocol (ThermoFisher). Briefly, we combined 250 μL from stool aliquots with 800 μL of Lysis Buffer, and vortexed the tube upside down for 10 s. Samples were placed on a shaker for 10 min at maximum speed (~2500 rpm). Then, 96-well plates were prepared according to the manufacturer’s protocol. Samples were centrifuged for 2 min at 14,000× *g* and 400 μL of the supernatant was transferred into row H of the prepared plate. We then added 520 μL of Binding Bead Mix ((500 μL Binding Buffer + 20 μL magnetic beads)/sample) to the samples and placed the plate in the KingFisher System. We ran the MagMAX Microbiome Ultra Nuclei program for RNA/DNA extraction and transferred the eluted samples into Eppendorf tubes stored at −80 °C. 

### 2.4. Library preparation and sequencing

Stool sample DNA (40 ng) was used to target 16S hypervariable regions V2-4 and V6-9 by PCR amplification. Libraries were prepared and purified for sequencing using the Ion 16S™ Metagenomics Kit (A26216, ThermoFisher) and Ion Express Barcode Adaptors 1–80 (ThermoFisher). Library sequencing analysis was performed using the Ion Reporter™ Software v5.18.4.0. Chimeric sequences were removed. Reads were mapped to Greenegenes v13.5 and MicroSEQ ID v3.0 as reference databases. Raw abundance values were subsampled to 10,000 reads per sample to control for inconsistent sample read numbers. Subsampling was performed using the species-level OTU table via the rrarefy function of the vegan package (version 2.5-6) [21]. All samples contained ≥10,000 raw read values, ensuring that none required removal from the analysis. Upstream taxonomic classification used current NCBI definitions via the taxize package (version 0.9.98) [22]. Shannon, Simpson, and Chao α-Diversity values were calculated, post-rarefaction, via the alpha function of the microbiome package (version 1.23.1) [23]. Results were tested for significance using a one-tailed *t*-test and *p* < 0.05. 

### 2.5. qPCR Assays

Stool sample RNA (40 ng) was converted to cDNA using SuperScript IV VILO Master Mix with ezDNase™ Enzyme (ThermoFisher). PCR primers were synthesized (ThermoFisher Scientific, Custom DNA Oligos Synthesis Services, Waltham, MA, USA) for the butyrate kinase (*BUK*) gene. DNA and cDNA yields (20 μL per sample) were subjected to quantitative PCR (qPCR; PerfeCTa SYBR Green FastMix, Quantabio, Beverly, MA, USA), in duplicate reactions, on 96-well plates, using a StepOnePlus Real-Time PCR instrument (ThermoFisher). Samples were incubated at 50 °C for 2 min, followed by initial denaturation at 95 °C for 2 min. Overall, 40 cycles of amplification were performed at 95 °C for 1 s and 60 °C for 20 s, and then they were held at 4 °C following the run. Cт (threshold cycle) results were normalized to BAC/16S amplification values, performed in the same plate, and the estimated percentage expression was calculated relative to BAC/16S using the delta-Cт method (equation: 2^(BAC Cт − BUK Cт)^ × 100). 

*BUK* qPCR primers were designed as previously described [24].

forward primer: 5′-TGCTGTWGTTGGWAGAGGYGGA-3′;reverse primer: 5′-GCAACIGCYTTTTGATTTAATGCATGG-3′.

Universal 16S qPCR primers (labeled “*BAC*”) were used as a proxy for total bacterial load. *BAC* primers were designed as previously described [25].

63F (forward primer): 5′-GCAGGCCTAACACATGCAAGTC-3′;355R (reverse primer): 5′-CTGCTGCCTCCCGTAGGAGT-3′.

miRNA was extracted from plasma samples according to the protocols for the TaqMan Adv miRNA cDNA synthesis kit (ThermoFisher) and the MagMAX mirVana Total RNA Isolation Kit (ThermoFisher) using the KingFisher DUO Prime automated extraction system. miRNA qPCR assays were performed in triplicate according to the TaqMan Adv miRNA protocol using the available human-specific TaqMan gene expression assays (ThermoFisher). The assays (and catalog numbers) used were miR-375-3p (#478074), miR-361-5p (#478056), miR-125b-5p (#477885), miR-132-3p (#477900), miR-134-5p (#477901), and miR-134-3p (#478408). Using the StepOnePlus instrument described above, qPCR conditions were 95 °C for 20 s, followed by 40 cycles of 95 °C for 1 s and 60 °C for 20 s. Once complete, samples were held at 4 °C. Cт results were normalized to miR-361 5p amplification as a loading control [26], performed in the same assay, and the estimated percentage expression was calculated relative to miR-361 using the delta-Cт method. Results were compared using an unpaired 2-tailed t test.

### 2.6. Luminex™ Assay

We designed a custom ProcartaPlex Luminex™ 14plex Assay (ThermoFisher), including beads for IL-1b, IL-4, IL-5, IL-6, IL-8, IL-10, IL-12p70, IL-17A IFN-γ, BDNF, TNF-α, PDGF, MIP-1*β*, and VEGF-A. We prepared the 96-well plate as described in the manufacturer’s protocol, except no plasma samples were diluted and all samples were centrifuged for 5 min at 1000× *g* before being added to the plate. The plate was read by a Luminex 200™ System using settings for MagPlex™ beads, as described in the ProcartaPlex Luminex™ Multiplex Immunoassay kit protocol. 

### 2.7. Ketone Body ELISA Assay

Pre- and post-KD plasma samples were checked for diet compliance using the Ketone Body Assay Kit (ab272541, Abcam, Cambridge, United Kingdom) according to the manufacturer’s protocol. The acetoacetate (AcAc) and *β*-hydroxybutyrate (BOH) assays were performed simultaneously, in duplicate, and read at 340 nm on a SpectraMax ABS plate reader (Molecular Devices, San Jose, CA, USA). Results were compared using a one-way ANOVA test.

### 2.8. Data Analysis

Data from Luminex, qPCR, and ELISA were analyzed and visualized using GraphPad Prism 9 (GraphPad Software, Boston, MA, USA). All pre-KD samples were compared to corresponding post-KD samples via paired one-tailed parametric t tests, unless otherwise indicated. Comparisons with a *p* value less than 0.05 were considered significant. 16S sequencing data were analyzed via computer pipeline as described above, and significance was measured at *p* < 0.05 without correction.

## 3. Results

### 3.1. Impacts of the KD on the gut microbiome in ASD children

Participants adhered to a modified KD for 4 months, and provided stool (*n* = 7) and blood (*n* = 11) samples before and after the diet (Figure 1a), which were used for analysis (Figure 1b,c). To biologically validate the physical status of ketosis, we measured plasma levels of ketone bodies using ELISA. As expected, we observed significantly increased concentrations of the ketone bodies acetoacetic acid (by 0.97 nM ± 0.66; *p* = 0.02) and hydroxybutyric acid (by 0.63 nM ± 0.38; *p* = 0.02) following the KD, as measured by a two-tailed one sample *t*-test (Table 1).

From the 16S-based sequencing data of gut microbial DNA from seven pairs of stool samples, we observed alterations in the phylum-level composition (Figure 2a) and a statistically significant increase in the α-diversity (independently measured by the Shannon and Simpson indices) of the gut microbiome in ASD children following the KD (Figure 2b,c). These changes were observed at the family, genus, and species levels.

Overall, the KD did not exhibit significant changes in phylum-level abundance, which could be attributed to the small sample size and individual microbial variability. However, at a higher resolution into lower taxonomic levels, significant increases were observed in *Lactobacillales* (*p* = 0.018 ± 0.005), while decreases were found in Bacteroidaceae (*p* = 0.030 ± 0.005), *Oscillospiraceae* (*p* = 0.040 ± 0.003), *Bacteroides* (*p* = 0.030 ± 0.048), *Ruminococcus* (*p* = 0.047 ± 0.003) genera, and *Clostridium cocleatum* (*p* = 0.041 ± 0.002) and *Ruminococcus gnavus* (*p* = 0.034 ± 0.002) species (Figure 2d–j).

Since previous analysis of this cohort has demonstrated a significant change in *β*-hydroxybutyrate in the blood [18], we examined stool samples for a corresponding increase in butyrate production capacity. Based on the qPCR of stool sample DNA and RNA, the abundance of the gene for *BUK*, and its expression levels, significantly increased after the KD (Figure 3). 

### 3.2. Impact of the KD on Inflammatory States in Children with ASD

We next examined changes to cytokines previously described as abnormally expressed in ASD patients: IL-1*β*, IL-4, IL-5, IL-6,IL-8, IL-10, IL-12p70, IL-17A, IFN-γ, BDNF, TNF-α, PDGF, MIP-1*β*, and VEGF-A [5,6,27,28,29]. We observed a general trend of a decreasing expression of pro-inflammatory cytokines in the plasma of 11 ASD patients following the KD, with significant differences observed in IL-1*β* and IL-12p70 levels (Table 2). Although the anti-inflammatory cytokine IL-10 increased after the KD, the mean difference was not statistically significant, likely due to the limited sample size (Table 2). Interestingly, we observed a significant decrease in plasma levels of BDNF (Table 2), an important factor involved in neuroinflammation in the central nervous system previously implicated in ASD [30].

### 3.3. KD-Induced Alterations to BDNF-Associated miRNAs

Given our observation of significantly decreased plasma levels of BDNF, we investigated the expression of four miRNAs previously linked to regulating BDNF activity in the brain: miR-134 5p [31], miR-132 3p [32], miR-125b 5p [33], and miR-375 3p [34], described in Table 3. Using targeted qPCR assays to measure these miRNAs from plasma samples, we observed significantly decreased levels of miR-134 (*p* = 0.008) and miR-132 (*p* = 0.027) following the KD (Figure 4b,d). The levels of miR-125b, however, remained unaltered after the KD compared to before it (Figure 4a, *p* = 0.238), while the levels of miR-375 appeared significantly elevated (Figure 4c, *p* = 0.044). 

## 4. Discussion

Previous reports from the authors on this cohort described improved social interactions [13] that correlated with increased plasma levels of *β*-hydroxybutyrate following a modified KD [18]. Although KD-induced ketosis and the elevation of circulating ketone bodies may directly account for modifying brain activity via changes to metabolic pathways and thereby explain improvements to sociability in ASD patients, our findings show additional effects of the KD that might impact brain activity. We observed significant alterations to the gut microbiome, inflammation, and circulating BDNF-associated miRNAs. These results are consistent with previous research suggesting that a KD improves social behavior in ASD via decreasing brain inflammation and ameliorating metabolic function, including altering the gut microbiota [11]. 

### 4.1. KD Promotes Taxonomic Richness and Evenness of the Gut Microbiome

In this pilot study, the KD led to a significant increase in gut microbiome biodiversity and composition at the family, genus, and species levels. The Shannon and Simpson indices showed greater evenness in population distribution and increased species richness. However, the Chao Index did not exhibit a significant change, possibly due to limited alterations in the abundance of rarer gut taxa [44]. In summary, the ketogenic diet (KD) appeared to positively influence the gut microbiota by promoting a more even distribution and increased richness. These findings generally suggest an improvement in gut health and a reduction in the pathogenicity of gut microbes.

### 4.2. Increased Butyrate Metabolism Is Associated with Reduced Inflammation

We assessed changes in butyrate concentration indirectly by investigating *BUK* DNA and RNA levels in the stool of participants prior to and after the KD. By measuring the relative abundance of *BUK* DNA and mRNA in the samples, we extrapolated relative microbiome-derived butyrate abundance. Previous metabolomic analysis of these samples observed low levels of *β*-hydroxybutyrate in ASD patients that were then elevated after the KD [18], further supporting evidence that gut microbiome changes affect blood metabolites. Blood levels of *β*-hydroxybutyrate have previously been shown to be associated with decreased levels of pro-inflammatory cytokines [45] and neuroinflammation [16]. These prior observations support our findings that the KD induces a significant shift toward an anti-inflammatory state in the participants, which might also contribute to the behavioral changes observed given prior links between inflammation and ASD symptomology [7].

### 4.3. KD-Induced Changes to BDNF-Associated miRNAs

Using Luminex assays, we further examined the activity of the neurotrophic factor BDNF. BDNF is found both in the brain and the periphery, where it regulates neurogenesis, glycogenesis, and synaptogenesis; provides neuroprotection; and controls short- and long-acting synaptic interactions in memory and cognition. Although its relationship with ASD symptoms is controversial, abnormal BDNF expression has been consistently found in ASD blood [30] and its activity influences and is influenced by epigenetic factors such as miRNAs that can alter inflammatory profiles [46]. Additionally, BDNF is known to interact with butyrate derived from the gut to modulate neuroinflammation and cell migration in the brain [47].

MicroRNAs (miRNAs) are small, endogenous noncoding RNAs that post-transcriptionally regulate gene expression, and have increasingly been found to be important in brain function and development [48]. By altering neuroinflammatory profiles, miRNAs may also play a role in modifying ASD conditions [49]. Thus, we tested four BDNF-associated miRNAs involved in neuronal development and activity: (1) miR134, downregulated by BDNF and expressed almost exclusively in the brain, is a negative regulator of dendrite width in mature neurons, through its silencing of *Limk1*. However, *Limk1* silencing can be relieved by the administration of BDNF, indicating a role of anti-inflammatory factors on the dendrite size [31]; miR-134 has also been confirmed to downregulate a related neurotropic factor, cerebral dopamine neurotrophic factor (CDNF) [36], as well as BDNF itself and cAMP-response-element binding protein (CREB-1) [35]; (2) miR132, found in the brain and other tissues [38], upregulates glutamate receptors [37], and is repressed by the downregulation of BDNF, with both implicated in disorders such as ASD and depression [37]. This miRNA has also been linked to pro-inflammatory activation [41]; (3) miR-125b controls spontaneous and BDNF-induced neuronal differentiation and growth [33], while also associating with increased inflammation [40]; and (4) miR-375 significantly regulates the differentiation and maturation of neurons in early development by maintaining stem-cell-like phenotypes. miR-375 has been shown to inhibit BDNF-promoted neuronal outgrowth [34]. It also plays a role in inflammation through cystatin SN (CST1) [42].

Our examination of the four BDNF-associated miRNAs consistently aligns with our findings of reduced plasma BDNF levels subsequent to the KD. Given BDNF’s recognized anti-inflammatory properties, its brain-specific nature, and its crucial role in neurodevelopment, the observed decrease post-KD might appear paradoxical, hinting at intricate associations with inflammation. Complicating matters, reports suggest abnormal BDNF elevation in ASD [30], implying that a lowered level could signify a shift towards “normal” concentrations. We propose that decreased BDNF expression in the periphery might stem from heightened sequestration in the brain, consequently diminishing peripheral blood concentrations. The observed miRNA expression alterations conform to established patterns from prior studies (see Figure 5), with butyrate itself exerting an influence on their expression, thus providing a connection between gut butyrate production and neuroinflammation via BDNF. While these initial findings implicate these miRNAs’ involvement, further investigation is imperative to elucidate their impact on ASD behavioral outcomes related to the KD, thereby establishing their potential as biomarkers.

## 5. Conclusions

Taken together, results from this interventional pilot study suggest an alternative mechanism through which a KD may influence behavioral symptoms in children with ASD, beyond the direct effect of ketones on metabolic activity in the brain (Figure 5). We showed that a KD increased gut microbial diversity, in addition to increasing the production of butyrate. Once metabolized in the gut, butyrate can enter the blood and the brain to promote an anti-inflammatory state directly, through action on cytokines such as IL-1*β* [56,57], or indirectly, through interactions with receptors such as TLR2 [50], neurotransmitters (e.g., acetylcholinesterase) [55], or miRNAs such as those associated with BDNF described above [52]. In addition to further supporting a correlation between inflammation and ASD, our interventional pilot study suggests that the gut microbiome can affect the expression of epigenetic factors (e.g., miRNAs) that might indirectly impact brain activity, adding to our understanding of the gut–brain axis and the mechanism by which a KD may impact ASD phenotypes.

Our preliminary findings of significant KD-related changes to the microbiome and neuroinflammatory biomarkers warrant further study in a larger cohort. Such studies should be powered to investigate age- and sex-related effects of the KD on sociability as well as on the microbiome and, importantly, may examine the persistence of these effects among children with ASD. Additionally, other gut microbiome metabolites, such as tryptophan metabolites and neurotransmitters [60], may contribute to the impact of the KD in children with ASD and present further avenues for future study. While preliminary, our findings highlight the need to better understand the effects of KD-related impacts on brain activity and behavior, supporting the potential relevance of butyrate production and inflammation in contributing to these effects via the gut microbiome.

## Figures and Tables

**Figure 1 nutrients-16-01401-f001:**
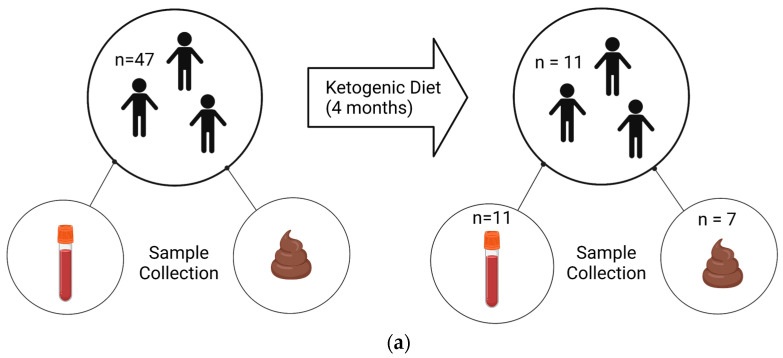
Experimental design and samples. Patients with an autism diagnosis were recruited from Oahu, Hawaii. (**a**) Blood and fecal samples were collected before and after 4 months on a KD. Samples were collected from 47 patients at the commencement of the study. At the end, 11 patients returned for follow-up, resulting in 11 blood samples and 7 stool samples available for analysis. (**b**) Non-fasting blood was collected between 9 and 10 am, and plasma was isolated. Subsequently, plasma was used in a ketone body assay and in a 14-plex cytokine Luminex assay. (**c**) Stools were collected via at-home collection kits, and then DNA and RNA were isolated and used for 16S sequencing and BUK/BAC qPCR assays. Created using Biorender.com.

**Figure 2 nutrients-16-01401-f002:**
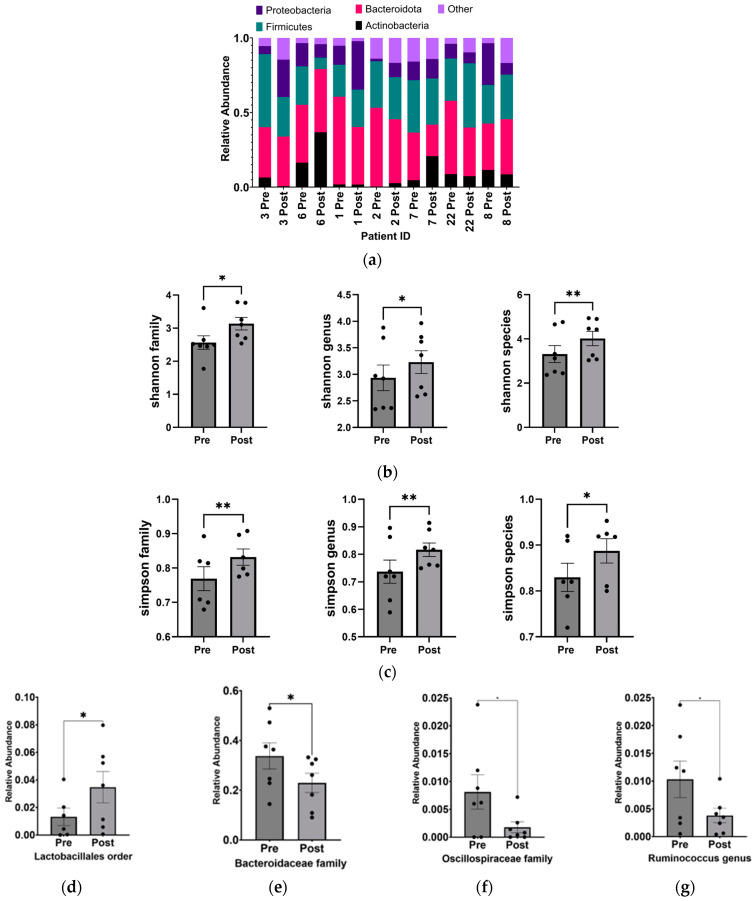
Increase in alpha diversity and changes to the abundance of specific gut microbiota after KD. (**a**) Relative abundance of phyla before and after KD in the same individual as measured by 16S sequencing. “Other” includes *Cyanobacteria*, *Fusobacteria*, *Lentisphaerae*, *Nitrospinae*, *Synergistetes*, *Tenericutes*, *Verrucomicrobia*, and Unclassified. Changes in microbial diversity before and after the KD based on 16S sequencing were measured using the (**b**) Shannon and (**c**) Simpson indices at the level of family, genus, and species. Bacteria found to be significantly different after KD are (**d**) *Lactobacillales* order (*p* = 0.02), (**e**) *Bacteroidaceae* family (*p* = 0.03), (**f**) *Oscillospiraceae* family (*p* = 0.04), (**g**) *Ruminococcus* genus (*p* = 0.04), (**h**) Bacteroides genus (*p* = 0.03), (**i**) *Ruminococcus gnavus* (*p* = 0.03), and (**j**) *Clostridium cocleatum* (*p* = 0.04). * *p* < 0.05, ** *p* < 0.001.

**Figure 3 nutrients-16-01401-f003:**
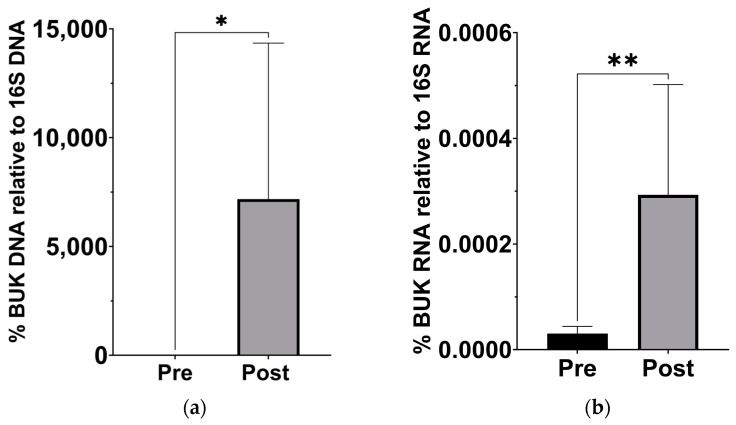
Increased levels of butyrate kinase gene and expression in gut microbiota after the KD. (**a**) DNA levels of BUK increased relative to 16S levels (*p* = 0.039; Wilcoxon matched-pairs signed rank t test, *n* = 7). (**b**) cDNA levels of BUK increased relative to 16S levels (*p* = 0.007; Wilcoxon matched-pairs signed rank t test, *n* = 7). * *p* < 0.05, ** *p* < 0.001.

**Figure 4 nutrients-16-01401-f004:**
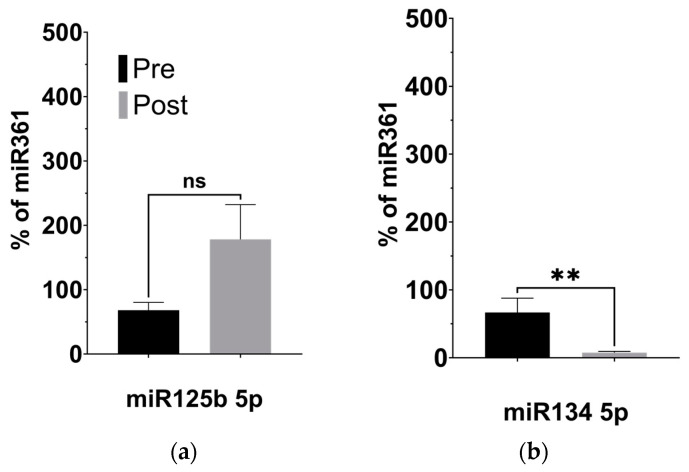
KD is associated with changes to the expression of BDNF-associated miRNAs. (**a**) miR125b increased after the KD relative to control miRNA361-5p, but not significantly. (**b**) miR134 5p decreased relative to control (*p* = 0.008; two-tailed unpaired *t*-test). (**c**) miR375 3p increased relative to control (*p* = 0.044; two-tailed unpaired *t*-test). (**d**) miR132 3p decreased relative to control miRNA361-5p (*p* = 0.027; two-tailed unpaired *t*-test). * *p* < 0.05, ** *p* < 0.001.

**Figure 5 nutrients-16-01401-f005:**
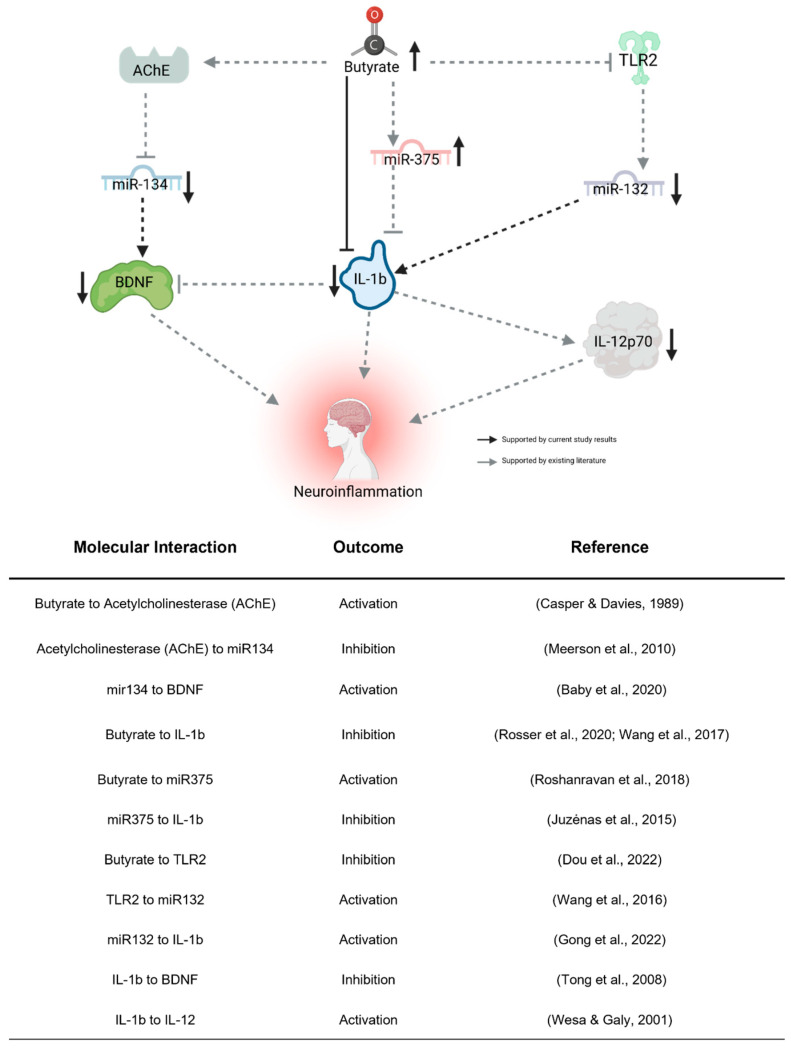
Proposed indirect pathway of KD-induced changes to neuroinflammation. Butyrate may downregulate miR-132 through the inhibition of TLR2 [50]. miR-132 promotes neuroinflammation by activating IL-1b [51]. Butyrate directly regulates miR-375 [52], which inhibits IL-1b [53]. Butyrate regulates miR-134 by activating its inhibitor, AChE [54,55]. miR-132 activates the neuroinflammatory effects of BDNF [35]. IL-1b, activated by miR-132 [51] and inhibited by miR-375 [53] and butyrate [56,57], influences neuroinflammation directly, through the inhibition of BDNF [58] and activation of IL-12p70 [59]. Created using Biorender.com (accessed 26 April 2023).

**Table 1 nutrients-16-01401-t001:** Summary of study participants.

**Characteristics**	***n* = 7**			
Age (years):	13.4 ± 3.8	Median	14	
		Range	7–19	
			**(%)**	
Gender:	Female	1	(14.3)	
	Male	6	(85.7)	
**Ketone Bodies**	**Pre-KD** **(Average ± SEM)**	**Post-KD** **(Average ± SEM)**	**Average Change in Plasma Concentration (nM)**	***p*-value**
Acetoacetic Acid	−0.39 ± 0.58	0.34 ± 0.63	0.98 ± 0.66	0.021
Hydroxybutyric Acid	0.62 ± 1.15	1.41 ± 1.63	0.64 ± 0.38	0.028

**Table 2 nutrients-16-01401-t002:** Average expression levels of ASD-associated cytokines before and after the KD.

Biomarker	Pre-KD (Average ± SEM)	Post-KD (Average ± SEM)	*p*-Value
IL-1b	1.03 ± 0.48	0.77 ± 0.39	0.04
IL-4	4.16 ± 0.12	4.03 ± 0.08	0.20
IL-5	12.37 ± 6.10	7.62 ± 4.62	0.06
IL-6	8.28 ± 0.73	8.28 ± 1.45	0.50
IL-10	0.28 ± 0.09	0.41 ± 0.25	0.14
IL-12p70	3.13 ± 2.05	1.00 ± 0.58	0.02
IL-17a	1.05 ± 0.41	0.84 ± 0.29	0.25
IFN-g	2.62 ± 0.48	2.37 ± 0.35	0.26
TNF-a	3.46 ± 1.86	2.93 ± 2.19	0.18
BDNF	3.32 ± 2.45	0.82 ± 0.29	0.02
PDGF-BB	23.77 ± 21.58	24.84 ± 23.76	0.01
VEGF-A	66.50 ± 28.75	57.87 ± 19.99	0.56

**Table 3 nutrients-16-01401-t003:** Validated and predicted targets of BDNF-associated miRNAs.

miRNA	Validated Targets	Predicted Targets ^1^
miR-134-5p	Limk1 [31]	DLG2, Neurod2, NDE1, GBX2
	CREB-1, BDNF [35]	
	CDNF [36]	
miR-132-3p	NR2A, Nr2B, GluR1 [37]	NREP, SLC6A1, NOVA1, MAF, SLC6A3, LRRTM3, NRCAM
	cAMP [38]	
	PTEN [39]	
miR-125b-5p	4E-BP1 [40]	MAF, GDNF, ELAVL4, NCAN
	TNF-α [41]	
miR-375-3p	PDK1, HuD [34]	ELAVL4, PACSIN1, ELAVL3, NBEA
	HNF1B, KLF4, CST1 [42]	

^1^ All predicted targets are based on in silico predictions by TargetScan [43].

## Data Availability

All data used for this project will be available, de-identified, when approved by the University of Hawaii Institutional Review Board, upon reasonable request to the corresponding author. The gut microbiome data presented in this study are deposited in the figshare repository as of 5 February 2023, accessible at: https://doi.org/10.6084/m9.figshare.22735193 (accessed on 5 February 2023).

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
