# Peer review of "Ketogenic Diet Induced Shifts in the Gut Microbiome Associate with Changes to Inflammatory Cytokines and Brain-Related miRNAs in Children with Autism Spectrum Disorder"

_nutrients, 2024, doi:10.3390/nu16101401_

Round 1
Reviewer 1 Report
Comments and Suggestions for Authors
The manuscript entitled “Ketogenic diet-induced shift in the gut microbiome associate with changes to inflammation and circulating brain-related miRNAs in children with autism spectrum disorder” by Allan and coworkers focused on the assessment of microbiota, inflammation markers, and microRNAs of seven children following a KD for 4 months.
This is a well-organized study but the concept of the study is not novel and the topic itself is important, however, the lack of clear rationale and mechanistic assessments reduces the scientific value of the paper.
According to a meta-analysis by Yu et al. 2022 (doi: 10.3389/fneur.2022.844117)
there are the results suggest the effectiveness of dietary therapy for ASD, but the small sample size of randomized controlled trials limits them. I agree with the Authors cited above that more well-designed, high-quality clinical trials are needed to validate the above conclusions. In the present study, the Authors showed a molecular analysis of a very small, and, in my opinion, not representative group (1 female and 6 male) which is known to be informative for the bigger cohorts (for example miRNAs). I agree with the Authors of the reviewed study that their findings warrant further study in a larger cohort. It is a matter of speculation to use a paired t-test for n=7. Such small size of samples should be analyzed with nonparametric tests. Please include in the revised version more details of KD composition (supplementary materials).
The authors did not show any ASD Assessment Scale/Screening Questionnaire or any psychological assessment but they concluded that KD may positively influence ASD sociability. A correlation between the effects of KD (molecular) and the improvement of sociability should be added to the study. Finally, I would suggest changing the title because the KD changed only two cytokines (changes in inflammation suggest broader changes).
Author Response
Reviewer 1:
Comments and Suggestions for Authors
The manuscript entitled “Ketogenic diet-induced shift in the gut microbiome associate with changes to inflammation and circulating brain-related miRNAs in children with autism spectrum disorder” by Allan and coworkers focused on the assessment of microbiota, inflammation markers, and microRNAs of seven children following a KD for 4 months.
This is a well-organized study but the concept of the study is not novel and the topic itself is important, however, the lack of clear rationale and mechanistic assessments reduces the scientific value of the paper.
According to a meta-analysis by Yu et al. 2022 (doi: 10.3389/fneur.2022.844117)
there are the results suggest the effectiveness of dietary therapy for ASD, but the small sample size of randomized controlled trials limits them. I agree with the Authors cited above that more well-designed, high-quality clinical trials are needed to validate the above conclusions. In the present study, the Authors showed a molecular analysis of a very small, and, in my opinion, not representative group (1 female and 6 male) which is known to be informative for the bigger cohorts (for example miRNAs). I agree with the Authors of the reviewed study that their findings warrant further study in a larger cohort. It is a matter of speculation to use a paired t-test for n=7. Such small size of samples should be analyzed with nonparametric tests.
Answer: Thank you for your feedback. We agree that the sample size is small, limiting our interpretation. However, few studies are able to pair stool and blood samples from children, especially with ASD, longitudinally and indeed this is difficult given the high attrition we observed. Despite this challenge, we present our results as preliminary and part of our pilot study, which warrant further investigation in a larger study.
Regarding the statistical tests, we felt that the Luminex assay was sensitive enough to assume a normal distribution as other studies from our group and that of others have shown, and thus used parametric tests. The miRNA qPCR results were the same regardless of if the test was parametric or non-parametric, but similar due to its high sensitivity, we considered presenting a parametric test more appropriate. The ketone body assay results were not formatted in a way that it was possible to perform a t test, and so were analyzed using a one-way ANOVA. Regarding the paired testing, one of the reasons our sample size is so small is because we limited our analysis to individuals who provided samples before and after the diet. Thus, all these samples are in fact paired and we feel that performing a paired test was appropriate and accurate despite the low sample size. We clarified the rationale of these tests in the methods section.
Please include in the revised version more details of KD composition (supplementary materials).
Answer: Thank you for the suggestion about the KD modifications. The detailed description of the modified KD was provided in our previous paper, cited in the methods. Since your feedback indicated a need for further detail, we included a brief summary of the modifications performed as described in (Lee 2018) in the methods section. Since these data and rationale were previously described in a paper that included all trial participants, we felt it more appropriate to direct further inquiry to the original publication.
The authors did not show any ASD Assessment Scale/Screening Questionnaire or any psychological assessment but they concluded that KD may positively influence ASD sociability. A correlation between the effects of KD (molecular) and the improvement of sociability should be added to the study.
Answer: We appreciate your concerns. As with the KD composition, the ADOS 2 scores for this cohort were reported in (Lee, 2018). Since these data are previously published and the underlying raw data is not available to us (it belongs to the hospital, which is not involved in this current paper), we can only provide citations to the results, not the actual data. We have modified the manuscript to emphasize that the ADOS 2 scores can be found in our previous publication and that they reflect the results from the same cohort.
Finally, I would suggest changing the title because the KD changed only two cytokines (changes in inflammation suggest broader changes).
Answer: Thank you for your suggestion. We have changed the title to “cytokines” instead of “inflammation”.

Reviewer 2 Report
Comments and Suggestions for Authors
To show the effects of ketogenic diet (KD) on autism spectrum disorder (ASD), authors studied in seven children with ASD the alterations of the gut microbiome (stool) as well as proinflammatory cytokines and brain-derived neurotrophic factor (BDNF)-associated microRNAs (blood), following KD for 4 months, and found their changes suggesting reduction of inflammation, improvement of gut microbiome composition and impact on the BDNF pathway.
This study is well designed and conducted. The findings are well presented and discussed. The limitation derives mostly from the small sample size.
1. Regarding participant enrollment, authors ascribed the small number of subjects to “personnel resources” (Line 80), however, it could have led to sampling bias. Authors should at least show the clinical improvement in ASD symptoms of the 7 (or 11?, according to Figure 1a) patients.
2. Which is the sample size for the blood studies (Sections 3.2 and 3.3), n = 7, 0r n = 11?
3. In the legend to Figure 2, the initial c of clostridium cocleatum (Line 212) should be in upper case.
Author Response
To show the effects of ketogenic diet (KD) on autism spectrum disorder (ASD), authors studied in seven children with ASD the alterations of the gut microbiome (stool) as well as proinflammatory cytokines and brain-derived neurotrophic factor (BDNF)-associated microRNAs (blood), following KD for 4 months, and found their changes suggesting reduction of inflammation, improvement of gut microbiome composition and impact on the BDNF pathway.
This study is well designed and conducted. The findings are well presented and discussed. The limitation derives mostly from the small sample size.
- Regarding participant enrollment, authors ascribed the small number of subjects to “personnel resources” (Line 80), however, it could have led to sampling bias. Authors should at least show the clinical improvement in ASD symptoms of the 7 (or 11?, according to Figure 1a) patients.
Answer: Thank you for your concern. We agree that our small sample size probably led to sampling bias, and we regret our inability to complete the trial as originally designed. A larger sample size would be needed to confirm or deny any of our observations in any kind of definitive manner. This would be a source for further study.
Clinical improvement in ASD symptoms were tested by our partners at Shriner’s Hospital, and reported previously in (Lee, 2018). Since these data are previously published and the underlying raw data is not available to us (it belongs to the hospital, which is not involved in this current paper), we can only provide citations to the results, not the actual data. We have modified the manuscript to emphasize that the ADOS 2 scores can be found in our previous publication and that they reflect the results from the same cohort.
We appreciate your confusion regarding the actual sample size. Since stool and blood samples were collected separately (the stool collection was a take-home kit), some individuals who provided blood samples did not return their stool samples. Thus, while we have 11 blood samples, we only have 7 stool samples. We made some minor changes to the text of the manuscript to make this clearer.
- Which is the sample size for the blood studies (Sections 3.2 and 3.3), n = 7, 0r n = 11?
Answer: The blood studies had a sample size of 11. We have made changes to the text to hopefully clear up this confusion.
- In the legend to Figure 2, the initial c of clostridium cocleatum (Line 212) should be in upper case.
Answer: Thank you for pointing this out. We have made the suggested change.
